# Improving the pragmatic usefulness of the scoring matrix for the Consolidated Framework for Implementation Research (CFIR). A proposal for a more frequency-based approach: The CFIR-f

George Economidis[1]*, Anne-Marie Eades[2]ʘ, Anthony Shakeshaft[1,3]ʘ, Sara Farnbach[1]ʘ

**1** Faculty of Medicine, National Drug and Alcohol Research Centre, University of New South Wales, Sydney, New South Wales, Australia, **2** Faculty of Health Sciences, Curtin School of Allied Health, Curtin University, Perth, Western Australia, Australia, **3** Faculty of Health and Behavioural Sciences, Poche Centre for Indigenous Health Services Research, University of Queensland, Brisbane, Queensland, Australia

ʘ These authors contributed equally to this work.
* g.economidis@unsw.edu.au

**Data Availability Statement:** All relevant data are within the manuscript and its Supporting Information files.

## Abstract

The Consolidated Framework for Implementation Research (CFIR) is a well-established framework for systematically identifying key factors influencing the implementation of programs. To enhance the classification of existing CFIR definitions, as well as its +2/-2 scoring system, this study incorporated the views of relevant experts to: i) improve *how* themes are scored; and ii) utilise more information regarding the *frequency* with which themes are identified. This structured, frequency-based approach to the CFIR's scoring process has been trialled as the CFIR-frequency (CFIR-f). Researchers thematically analysed semi-structured interview data from four groups of policy and program experts (N = 24) delivering two family-based therapies in New South Wales (NSW), Australia. Themes identified by less than 50% were excluded from further analysis. Themes identified by 50% or more of expert participants in the four groups were classified as enablers or barriers using clearly defined criteria. Each theme was allocated a score according to how many experts identified it as an enabler or barrier, and then mapped back onto the latest adaptation of the CFIR comprising 67 constructs. The CFIR-f successfully determined three enablers of, and six barriers to, implementation. Enablers included the family-based therapy programs, therapist training and participant monitoring systems. Barriers included referral, data collection and staffing difficulties, NSW adaptation issues and the suitability and safety of the programs for Aboriginal families. The same enablers and barriers were identified using both the CFIR-f and the original scoring approach, and the identified themes were successfully mapped to almost all CFIR constructs (65/67). This paper proposes a more frequency-based approach to CFIR's scoring process (the CFIR-f). By specifically utilising the frequency with which these barriers and enablers are identified, the CFIR-f engenders a list of ranked themes that service

**Funding:** GE was supported by an Australian Government Research Training Program (RTP) Scholarship via the University of New South Wales (UNSW), Sydney, Australia, and a Higher Degree Research scholarship from the National Drug and Alcohol Research Centre (NDARC), UNSW, Sydney. The evaluation of the MST-CAN program in New South Wales, Australia (2018-2020) was funded via a competitive tender process (FACS.17.266) by the NSW Government Department of Family and Community Services (now Department of Communities and Justice). Their Futures Matter and Department of Communities and Justice staff provided feedback on the evaluation plan, participated in interviews as key stakeholders in this evaluation, and reviewed the manuscript.

**Competing interests:** All authors have approved the manuscript for submission. One co-author, Anthony Shakeshaft, was the lead investigator on a competitive tender (FACS.17.266) awarded by the NSW Government Department of Family and Community Services (now Department of Communities and Justice) to evaluate the Functional Family Therapy – Child Welfare [FFT-CW®] and Multisystemic Therapy for Child Abuse and Neglect [MST-CAN®] programs in New South Wales, Australia (2018-2020). The lead author, George Economidis, was employed part-time at the National Drug and Alcohol Research Centre as a Project Coordinator for the FFT-CW® and MST-CAN program evaluation from May 2018 to August 2020. This does not alter our adherence to PLOS ONE policies on sharing data and materials.

providers and policymakers can use to inform their decisions about program modification and implementation.

## Introduction

Programs that simultaneously respond to harms that emanate from the individual, community and societal level are increasingly being utilised, both to optimise their effectiveness and improve the sustainability with which they are integrated into real-world service delivery [1, 2]. The increased complexity of the methods required to evaluate these multi-level, complex programs means that the cost of designing and implementing evaluations has also increased [3]. Identifying the enablers of, and barriers to, the delivery of multi-level programs at an early stage of their implementation is especially important because it is a particularly volatile stage in the program delivery process [3, 4]. Moreover, identifying the factors that influence the early implementation of a program creates the opportunity for the enablers to be leveraged and the barriers to be modified, to optimally tailor the program to the context in which it is being delivered before the processes that support its implementation become entrenched [5]. Taxonomic frameworks have been developed to standardise the systematic identification of enablers of, and barriers to, implementation [6, 7]. One such framework is the Consolidated Framework for Implementation Research (CFIR) [6].

The CFIR is a conceptual, systematic and meta-theoretical approach that is used to assess the uptake of a program against key factors that have been shown to influence the implementation of complex programs [8–10]. The CFIR has been used to assess the implementation of health-related services, such as weight management [7], hospital care [11], chronic disease [12] and pharmacy services [13], and the implementation of social services, such as the introduction of intensive family-based therapy programs in the Australian state of Victoria [4].

The CFIR is described in detail elsewhere [6]. In brief, evaluators have used the CFIR to measure the impact of a wide range of variables on the implementation of a program by using data generated by both qualitative and quantitative research methods. From a qualitative perspective, evaluators have used the CFIR as a framework, or 'checklist', for coding information collected using standardised qualitative techniques, such as focus group discussions and semi-structured interviews [10]. From a quantitative and mixed-methods perspective, evaluators have applied ratings, or rankings, to the CFIR's constructs based on the results of surveys or questionnaires [7, 14]. It is worth noting that most studies have only used the CFIR to inform their data analysis rather than their data collection [10].

The latest adaptation of the CFIR (CFIR 2.0) comprises 67 constructs that are organised into five domains: 1) innovation (e.g., the distinct features and components of the program); 2) outer setting (the external features of the program); 3) inner setting (the internal features of the program); 4) individuals (e.g., the personal traits, attributes and roles of staff responsible for program delivery); and 5) implementation (the change strategies adopted by the organisation) [15]. Evaluators then frequently apply a scoring system, whereby each theme is allocated a score from +2 to -2: themes rated +1 or +2 are classified as enablers; themes rated -1 or -2 are classified as barriers; and themes rated as 0 are classified as being neither an enabler nor a barrier to implementation [7, 16]. The more extreme scores (i.e., +2 and -2) are interpreted as more influential barriers and enablers, while +1 and -1 are interpreted as being generally influential. These categorised themes can then be mapped back onto the five CFIR domains.

Although this process is clearly defined and useful overall, its pragmatic value may be improved both in terms of *how* themes are scored and the *frequency* with which themes are

identified. In relation to *how* themes are scored, for example, a theme would be allocated a score of -1 (and consequently interpreted as a barrier with a generally negative influence) if it was only discussed "in passing" or at a "high level without examples" [7]. A potential weakness in this process is that the definitions of these two phrases "in passing" and "high level" are potentially too vague: they could be interpreted differently by various independent raters (a challenge to inter-rater reliability), or interpreted differently by the same independent rater in various contexts or at different points in time (a challenge to intra-rater reliability). These challenges to inter- and intra-rater reliability could create artificial variation in the results of a CFIR evaluation.

In relation to assessing the *frequency* with which themes are identified, a theme would be allocated a score of -2 or +2 (and consequently interpreted as a barrier or an enabler) if it meets the threshold of being ". . .discussed by at least two interviewees using explicit examples. . ." [7]. A potential weakness in this process is that a theme discussed by two interviewees is treated in the same way as a theme discussed by five or 10 interviewees. This process may subsequently miss the opportunity to identify themes according to the frequency with which they are raised by expert participants, rather than the current dichotomous categorisation with a cut-off of two interviewees. The pragmatic value in improving *how* themes are scored is that service providers and funders would have greater confidence in the reliability with which themes are classified as barriers or enablers. The pragmatic value in utilising more information regarding the *frequency* with which experts identify themes is that service providers and funders would obtain more actionable information to inform their decisions about how to improve programs, or discontinue their investment in them, because implementation strategies are based specifically on the number of participants who identified themes as enablers or barriers to implementation.

Consequently, the aim of this study is to utilise the opportunity of two family-based therapy programs being integrated into the routine delivery of child protection services in New South Wales (NSW), Australia, to propose a more frequency-based approach to the CFIR's classification and scoring process: the CFIR-frequency (CFIR-f).

## Methods

### Family-based therapy services in NSW

The CFIR framework was used in a mixed-methods process evaluation of the implementation of Functional Family Therapy (FFT-CW®) and Multisystemic Therapy for Child Abuse and Neglect (MST-CAN) into the routine delivery of child protection services in NSW, Australia [17]. These government-funded programs use a therapeutic, strengths-based and whole-of-family approach aimed at reducing the likelihood that children who are at high risk of maltreatment will be placed into out-of-home care. The CFIR framework was used to assess the fidelity and success with which FFT-CW® and MST-CAN were being implemented into the routine delivery of child protection services in NSW.

### Participants in the process evaluation

Twenty-four policy and program experts (defined as those actively engaged in the adaptation, planning, implementation and/or monitoring of either MST-CAN or FFT-CW®) were purposively recruited in August 2018 and allocated into one of four groups based on their role: i) model purveyors (five clinicians and/or program experts who own the intellectual property and licensing rights of both models); ii) intermediaries (seven domestic or international implementation support organisations who provided adherence and fidelity support to service providers); iii) representatives of the NSW Government's Their Futures Matter team responsible

for the implementation of MST-CAN and FFT-CW® in NSW (six staffers who oversaw the implementation and contractual arrangements for service providers to deliver both programs in NSW); and iv) staff of the NSW Government's Department of Communities and Justice (DCJ) (six staffers who commissioned and referred high-risk and eligible families to the programs). Considering both MST-CAN and FFT-CW® were implemented across NSW at the same time and for largely the same purpose, the CFIR was used to evaluate the implementation of both programs simultaneously, irrespective of the program in which participants were involved, noting that the program-specific enablers and barriers to the implementation of both programs in NSW have been reported elsewhere [18, 19]. Details about the roles of the policy and program experts, and the broader organisational network involved in the implementation of MST-CAN and FFT-CW® are provided elsewhere [17], as are details about the specific qualitative methods used to evaluate the uptake process for MST-CAN [18] and FFT-CW® [19].

## Identifying themes

Twenty-four in-depth, semi-structured interviews were undertaken between August and September 2018 (see [18, 19] for further information regarding the sampling methodology utilised, and for the interview questions asked of policy and program experts). Responses were deductively and inductively thematically analysed using a reflexive approach [20–22], which identified the 16 themes summarised in Table 2 (further details on how this thematic analysis was undertaken are provided elsewhere [18, 19]). Authors did not have access to information that could identify individual participants during or after data collection.

## Procedures for this analysis

A modified CFIR analysis process (the CFIR-f) is proposed, comprising four steps.

First, as with current practice using the CFIR, issues related to the implementation of a program are identified using an established qualitative method (e.g., semi-structured interviews), consisting of questions that align with the CFIR's 67 constructs. These issues are then transposed into themes, using standard thematic analysis.

Second, themes were excluded as being too infrequently identified if they were mentioned by less than 50% of participants. Introducing a cut-off of 50% aimed to concentrate the focus of the analysis on the themes that emerged from the issues identified by the majority of group participants, rather than including all themes. The aim of this step is to identify the barriers and enablers with the most expert consensus so that service providers and policymakers can have more confidence in their decisions about program modification and implementation.

Third, for the themes identified by at least 50% of participants in each group, the number of participants who identified it as a barrier or an enabler was recorded by the researchers, and that number was allocated to each barrier (up to -5 for model purveyors because there were five participants in this group, -7 for intermediaries, -6 for Their Futures Matter staff and -6 for DCJ staff) and enabler (up to +5 for model purveyors, +7 for intermediaries, +6 for Their Futures Matter staff and +6 for DCJ staff). In addition, a theme was classified as neither an enabler nor barrier to implementation (termed 'neutral' and scored 0) if at least one of two criteria were met: 1) the majority of group participants discussed the theme in neutral terms (e.g., being 'indifferent' or 'unclear' about the impact of the theme) or specifically indicated that the theme neither enhanced nor inhibited program implementation; or 2) an equal number of group participants discussed the theme as an enabler ***and*** barrier to program implementation in NSW (or, for groups with odd numbers of participants, if the number of group participants who discussed a theme as both an enabler and barrier differed by one person). These two criteria for neutrality are strongly aligned with the original CFIR criteria for neutrality (that there is

no evidence of positive or negative impact on implementation, or if a theme is discussed in both positive and negative terms) but differ by specifying the objective threshold of a majority of group participants (criterion 1) or an equal number of group participants (criterion 2).

Fourth, the score for each group was summed to give a total score for each theme, resulting in a ranked list of themes for relevant stakeholders to action. The logic is to provide services, funders and policymakers with a list of barriers and enablers determined by the frequency with which they are mentioned by participants across each of the four expert groups.

To illustrate the application of these four steps in the current study, the emergent theme *outcome measures and data collection (theme 2 in Table 2)* was identified by 14 interviewees across the four FFT-CW® and MST-CAN participant groups. The theme was scored as being of low relevance only in the model purveyor group (less than 50% of model purveyors identified this theme). Fifty percent or more of participants in both the intermediary and Their Futures Matter groups identified this theme and specified it as barrier, largely due to the administrative and logistical difficulties associated with obtaining and storing data. Considering four expert participants in the intermediary group and four expert participants in the Their Futures Matter group discussed this theme as a barrier, each group was allocated a score of -4 for this theme. Although more than 50% of participants in the DCJ group also identified this theme, it was rated as neutral (and scored 0) because equal numbers of group participants rated it as a potential enabler (e.g., noting the benefits of a meticulous and centralised database that is immediate and easily accessible) and a potential barrier (e.g., highlighting similar barriers to implementation discussed by intermediaries). As a result, the overall score for the theme, using the proposed modified for the CFIR-f, is -8.

The differences between the original CFIR approach and the CFIR-f are summarised in Table 1.

## Mapping themes onto the 67-construct CFIR

To identify the extent to which the themes identified by the CFIR-f were able to be mapped to the CFIR constructs, themes were independently mapped onto the 67 CFIR constructs by two researchers: authors GE and AME. The mapped constructs were compared and discussed until consensus was reached. In addition, the themes identified by the CFIR and the themes identified by the CFIR-f were compared to identify similarities and differences in the enablers and barriers to the implementation of FFT-CW® and MST-CAN generated by these two different scoring methods (see Table 3).

## Ethics approval

Ethics approval was obtained from the University of NSW's Human Research Ethics Committee (HC180375) and the NSW Aboriginal Health and Medical Research Council (1429/18).

## Results

### Identifying themes using the proposed adapted scoring method

Table 2 shows the 16 themes identified, ranked by both their CFIR score (ranging from -6 to +4) and their CFIR-f score (ranging from -11 to +7), and the number of participants in each of the four groups who identified each theme. Using the CFIR-f scoring method, 13 of the 16 emergent themes were identified by 50% or more of Their Futures Matter staff and could thus be scored, 11 were identified by 50% or more of the intermediaries, six were identified by 50% or more of the model purveyors, and five were identified by 50% or more of DCJ staff. This

**Table 1. Summary of the differences between the original CFIR and adapted CFIR-f approach.**

| Original CFIR approach | Adapted CFIR-f approach |
|---|---|
| 1) All themes included | 1) Only themes identified by at least 50% of participants allocated a score |
| 2) Themes rated as a generally negative influence/ barrier if discussed "in passing" or at a "high level without examples" (-1) or as a negative influence/ barrier if at least two interviewees discussed it as negatively impacting implementation (-2) | 2) Themes rated as barrier if the majority of participants (i.e., at least 50% or more of group participants) discussed the theme as a barrier. Score of barrier reflects number of participants in each group who identified the theme as such (e.g., up to -5 for model purveyors) |
| 3) Themes rated as a generally positive influence/ enabler if discussed "in passing" or at a "high level without examples" (+1) or as a positive influence/ enabler if at least two interviewees discussed it as positively impacting implementation (+2) | 3) Themes rated as enabler if the majority of participants (i.e., at least 50% or more of group participants) discussed the theme as an enabler. Score of enabler reflects number of participants in each group who identified the theme as such (e.g., up to +5 for model purveyors) |
| 4) Themes rated as neutral (0) if there is no evidence of positive or negative impact on implementation, or if discussed in both positive and negative terms | 4) Themes rated as neutral if 1) the majority of group participants discussed the theme in neutral terms (e.g., if they were 'indifferent' or 'unclear' as to the impact of the theme, or if they stressed the theme neither enhanced nor inhibited program implementation), or 2) an equal number of group participants discussed the theme as an enabler **and** barrier to program implementation in NSW, or if the number of group participants who discussed a theme as both an enabler and barrier differed by one person (in instances when group sizes total an odd number, like seven or nine). *These two criteria for neutrality are strongly aligned with the original CFIR criteria for neutrality (that there is no evidence of positive or negative impact on implementation, of if a theme is discussed in both positive and negative terms) but differ by specifying the objective threshold of a majority of group participants (criterion 1) or an equal number of group participants/group participants differed by one person (criterion 2)* |

resulted in the identification of three enablers, six barriers and seven neutral themes, as depicted in Table 2.

Of these six barriers, technical, logistical and referral challenges (scored as -11) was identified by all experts, (except DCJ staff). Workforce, staffing, resourcing and funding issues (scored as -7) was identified as a barrier by model purveyors and intermediaries, and as neither an enabler nor barrier by Their Futures Matter staff, and was not identified at all by DCJ staff. Outcome measures and data collection issues (scored as -8) was highlighted as a barrier by intermediaries and Their Futures Matter staff, whereas the eligibility criteria of the programs (scored as -3) was identified as a barrier by only Their Futures Matter staff. The adaptation challenges to the NSW context (scored as -4), as well as the evidence base and appropriateness of the programs for Aboriginal families (scored -5), were perceived as barriers only by inter- mediary organisations.

The three enablers included: the nature and structure of the FFT-CW® program (scored as +7), model purveyors' training and support of therapists (scored as +4), and the ability to monitor a family's progress in the programs in real-time (scored as +3). Of these, one was rated as an enabler by Their Futures Matter staff and intermediaries (the nature and structure of the FFT-CW® program (scored as +7)), one was rated as an enabler by Their Futures Matter staff only (the ability to monitor a family's progress (scored as +3)), and one was rated as an enabler by model purveyors only (role of model purveyors' training and support of therapists (scored as +4)). A full list of the key ideas underpinning each of the themes in Table 2 used to generate them as either enablers, barriers or neither, as well as supporting interviewee quotes, is detailed in S1 Table.

**Table 2. The 16 themes identified as enablers and barriers, and neither enablers or barriers, to the implementation of MST-CAN and FFT-CW® in NSW.**

| Themes (n = total number of participants who identified each theme) | Model purveyors (n = 5) | Intermediaries (n = 7) | TFM (n = 6) | DCJ (n = 6) | CFIR score^ |
|---|---|---|---|---|---|
| *Barriers* | | | | | |
| 1. Technical, logistical and referral challenges (n = 19)* | -3 <br> **-2** | -4 <br> **-2** | -4 <br> **-2** | <50%[a] <br> **0** | -11 <br> **-6** |
| 2. Outcome measures and data collection (n = 14) | <50% <br> **0** | -4 <br> **-2** | -4 <br> **-2** | 0 <br> **0** | -8 <br> **-4** |
| 3. Workforce, staffing, resourcing and funding issues (n = 13) | -3 <br> **-2** | -4 <br> **-2** | 0 <br> **0** | <50% <br> **0** | -7 <br> **-4** |
| 4. Evidence base and appropriateness for Aboriginal families (n = 16) | <50% <br> **0** | -5 <br> **-2** | 0 <br> **0** | <50% <br> **0** | -5 <br> **-2** |
| 5. Adaptation to NSW context (n = 16) | 0 <br> **0** | -4 <br> **-2** | 0 <br> **0** | <50% <br> **0** | -4 <br> **-2** |
| 6. Eligibility criteria of programs (n = 9) | <50% <br> **0** | <50% <br> **0** | -3 <br> **-2** | 0 <br> **0** | -3 <br> **-2** |
| *Neither enablers or barriers* | | | | | |
| 1. Relationship with service providers and other stakeholders (n = 15) | <50% <br> **0** | 0 <br> **0** | 0 <br> **0** | 0 <br> **0** | 0 <br> **0** |
| 2. Role of policy expert/nature of contact with the program (n = 13) | <50% <br> **0** | 0 <br> **0** | <50% <br> **0** | 0 <br> **0** | 0 <br> **0** |
| 3. Case management processes (n = 11) | 0 <br> **0** | 0 <br> **0** | <50% <br> **0** | <50% <br> **0** | 0 <br> **0** |
| 4. Implementation facilitators and barriers (n = 11) | <50% <br> **0** | 0 <br> **0** | 0 <br> **0** | 0 <br> **0** | 0 <br> **0** |
| 5. Role of intermediaries (n = 10) | <50% <br> **0** | 0 <br> **0** | 0 <br> **0** | <50% <br> **0** | 0 <br> **0** |
| 6. Procurement & contracting (n = 9) | <50% <br> **0** | <50% <br> **0** | 0 <br> **0** | <50% <br> **0** | 0 <br> **0** |
| 7. Pre-implementation phase procedures (n = 8) | 0 <br> **0** | <50% <br> **0** | 0 <br> **0** | <50% <br> **0** | 0 <br> **0** |
| *Enablers* | | | | | |
| 1. Client snapshot/real-time monitoring of families (n = 5) | <50% <br> **0** | <50% <br> **0** | +3 <br> **+2** | <50% <br> **0** | +3 <br> **+2** |
| 2. Model purveyor training and support (n = 6) | +4 <br> **+2** | <50% <br> **0** | <50% <br> **0** | <50% <br> **0** | +4 <br> **+2** |
| 3. Nature and structure of FFT-CW® program (n = 9) | <50% <br> **0** | +4 <br> **+2** | +3 <br> **+2** | <50% <br> **0** | +7 <br> **+4** |

(-2) = Most commonly identified theme as a barrier

(0) = Not strongly identified as a barrier or enabler themes

(2) = Most commonly identified theme as an enabler

a = Infrequently identified (i.e., a theme discussed by less than 50% of participants—e.g., <3 of 5 model purveyors)

* = Number of participants across the four groups who identified each theme

^ = Scores not in bold derived from the CFIR-f scoring system; scores in bold derived from the original CFIR scoring system

## Mapping themes onto the 67-construct CFIR framework

Table 3 shows that the themes generated by policy and program experts using the CFIR-f, as summarised in Table 2, were mapped to 65 of the 67 constructs encompassed in CFIR 2.0. There was 100% agreement between raters regarding which themes mapped to which of the 65 CFIR constructs. The two CFIR constructs to which no themes generated by the CFIR-f were able to be mapped were: 1) Innovation Trialability (i.e., piloting the program by delivering it on a much smaller scale prior to its wider implementation); and 2) Critical Incidents (i.e.,

**Table 3. Identified themes mapped to the original CFIR framework, using original CFIR scoring and proposed CFIR-f scoring.**

| Themes | CFIR domains/constructs to which each identified theme was mapped | Original CFIR score | Proposed CFIR-f score |
|---|---|---|---|
| *Barriers* | | | |
| 1. Technical, logistical and referral challenges | **1F)** Innovation Complexity<br>**3F)** Compatibility | -6 | -11 |
| 2. Outcome measures and data collection | **2Giii)** Performance Measurement Pressure<br>**3Aii)** Information Technology Infrastructure<br>**3Diiii)** Learning-Centredness | -4 | -8 |
| 3. Workforce, staffing, resourcing and funding issues | **2D)** Partnerships & Connections<br>**2G)** External Pressure<br>**2Gi)** Societal Pressure<br>**2Gii)** Market Pressure<br>**3A)** Structural Characteristics<br>**3Ai)** Physical Infrastructure<br>**3Aii)** Information Technology Infrastructure<br>**3Aiii)** Work Infrastructure<br>**3B)** Relational Connections<br>**3C)** Communications<br>**3D)** Culture<br>**3Di)** Human Equality-Centredness<br>**3Dii)** Recipient-Centredness<br>**3Diii)** Deliverer-Centredness<br>**3Div)** Learning-Centredness<br>**3G)** Relative priority<br>**3H)** Incentive Systems<br>**3I)** Mission Alignment<br>**3J)** Available Resources<br>**3Ji)** Funding<br>**3Jii)** Space<br>**3Jiii)** Materials & Equipment<br>**3K)** Access to Knowledge & Information<br>**4A)** High-Level Leaders<br>**4B)** Mid-Level Leaders<br>**4C)** Opinion Leaders<br>**4D)** Implementation Facilitators<br>**4E)** Implementation Leads<br>**4F)** Implementation Team Members<br>**4G)** Other Implementation Support<br>**4H)** Innovation Deliverers<br>**5A)** Teaming | -4 | -7 |
| 4. Evidence base and appropriateness for Aboriginal families | **1B)** Innovation Evidence Base<br>**ID)** Innovation Adaptability<br>**2B)** Local Conditions<br>**2E)** Policies & Laws<br>**2Gi)** Societal Pressure<br>**2E)** Tension for Change<br>**4I)** Innovation Recipients<br>**5B)** Assessing Needs<br>**5Bii)** Innovation Recipients<br>**5C)** Assessing Context<br>**5E)** Tailoring Strategies<br>**5F)** Engaging<br>**5Fii)** Innovation Recipients<br>**5H)** Reflecting & Evaluating<br>**5Hi)** Implementation<br>**5Hii)** Innovation<br>**5I)** Adapting | -2 | -5 |

(*Continued*)

**Table 3.** (Continued)

| Themes | CFIR domains/constructs to which each identified theme was mapped | Original CFIR score | Proposed CFIR-f score |
|---|---|---|---|
| 5. Adaptation to NSW context | **1A)** Innovation Source<br>**1D)** Innovation Adaptability<br>**1G)** Innovation Design<br>**1H)** Innovation Cost<br>**2B)** Local Attitudes<br>**2C)** Local Conditions<br>**2E)** Policies & Laws<br>**2d)** External policies and incentives<br>**3E)** Tension for Change<br>**3K)** Access to Knowledge and Information<br>**4A)** High-Level Leaders<br>**4B)** Mid-Level Leaders<br>**4C)** Opinion Leaders<br>**4D)** Implementation Facilitators<br>**4E)** Implementation Leads<br>**4F)** Implementation Team Members<br>**4G)** Other Implementation Support<br>**4H)** Innovation Deliverers<br>**5D)** Planning<br>**5G)** Doing<br>**5I)** Adapting | -2 | -4 |
| 6. Eligibility criteria of programs | **1F)** Innovation Complexity | -2 | -3 |
| *Neither enablers or barriers* | | | |
| 1. Relationship with service providers and other stakeholders | **1A)** Innovation Source<br>**2D)** Partnerships & Connections<br>**3B)** Relational Connections<br>**3C)** Communications<br>**3D)** Culture<br>**5A)** Teaming | 0 | 0 |
| 2. Role of policy expert/nature of contact with the program | **4C)** Opinion Leaders<br>**4D)** Implementation Facilitators<br>**4F)** Implementation Team Members<br>**4J)** Need*<br>**4K)** Capability*<br>**4L)** Opportunity*<br>**4M)** Motivation* | 0 | 0 |
| 3. Case management processes | **1G)** Innovation Design<br>**3Di)** Human Equality-Centredness<br>**3Dii)** Recipient-Centredness | 0 | 0 |
| 4. Implementation facilitators and barriers | **3E)** Tension for Change<br>**5D)** Planning<br>**5G)** Doing | 0 | 0 |
| 5. Role of intermediaries | **1D)** Innovation Adaptability<br>**3Dii)** Recipient-Centredness<br>**3Diii)** Deliverer-Centredness | 0 | 0 |
| 6. Procurement & contracting | **1H)** Innovation Cost<br>**2E)** Policies & Laws<br>**2F)** Financing<br>**2G)** External Pressure<br>**3E)** Tension for Change | 0 | 0 |
| 7. Pre-implementation phase procedures | **3E)** Tension for Change<br>**3K)** Access to Knowledge & Information<br>**5B)** Assessing Needs<br>**5C)** Assessing Context | 0 | 0 |
| *Enablers* | | | |

(*Continued*)

**Table 3.** (Continued)

| Themes | CFIR domains/constructs to which each identified theme was mapped | Original CFIR score | Proposed CFIR-f score |
|---|---|---|---|
| 1. Client snapshot/real-time monitoring of families | **4J)** Need*<br>**5Bi)** Innovation Deliverers<br>**5Fi)** Innovation Deliverers<br>**5H)** Reflecting & Evaluating | +2 | +3 |
| 2. Model purveyor training and support | **2K)** Access to Knowledge & Information<br>**4A)** High-level leaders<br>**4E)** Implementation Leads<br>**5Hi)** Implementation<br>**5Hii)** Innovation | +2 | +4 |
| 3. Nature and structure of FFT-CW® program | **1B)** Innovation Evidence Base<br>**1C)** Relative Advantage<br>**1G)** Innovation Design<br>**3G)** Relative Priority<br>**4J)** Need* | +4 | +7 |
| *CFIR constructs for which no themes were identified* | | | |
| | Innovation Trialability | n/a | n/a |
| | Critical Incidents | n/a | n/a |

*For *4) Individuals Domain*, project characteristic constructs ('Need', 'Capability', 'Opportunity' and 'Motivation') in *Characteristics* subdomain are all relevant to this evaluation, and as such, were labelled as a continuation of constructs in *Project Roles* subdomain (unlike the separate lettering format utilised in [15], the authors of which are responsible for the formulation of this latest CFIR framework adaptation).

significant delays in program implementation due to unforeseen, wide-scale interruptions) [15]. Definitions of each these CFIR constructs, and explanations for how FFT-CW® and MST-CAN align with each, is articulated in S2 Table.

In addition to summarising how the themes identified by the CFIR-f mapped back to the 67 CFIR 2.0 constructs, Table 3 shows three key features that highlight the potential of the CFIR-f scoring system proposed in this paper. First, it clearly shows that both the CFIR and CFIR-f scoring systems identify the same enabler themes (n = 3), the same barrier themes (n = 6) and the same neutral themes (n = 7). This replication of themes as being enablers, barriers or neutral provides reassurance of the concurrent validity of the CFIR-f scoring system: using it resulted in the themes being classified in exactly the same way as the original CFIR scoring, which can be regarded as the current gold standard CFIR scoring method [23].

Second, it highlights the practical value of the additional information that can be derived from the CFIR-f. Specifically, Table 3 shows that the original CFIR scores are only capable of generating an ordinal list of themes because the original scoring method only applies scores of -2 and +2 (where the theme was raised by two or more interviewees using explicit examples). For example, using the original scoring method, the theme *workforce, staffing, resourcing and funding issues (theme 3 in Tables 2 & 3)* was discussed with specific examples by at least two participants in the model purveyors group and at least two participants in the intermediaries group, meaning it scored -2 for each of these groups separately. This theme also scored a 0 for the Their Futures Matter and the DCJ groups because it was identified in neutral terms by participants in those groups. Consequently, this theme received a total CFIR score of -4. When applied to all themes, Table 3 shows this original CFIR scoring mechanism generates an ordinal list of themes that can only score 0 or +/- 2, 4, 6 and 8.

In contrast, the proposed CFIR-f scoring system generates a continuous range of scores for each theme (bounded by the number of participants). These scores can be interpreted as a list of ranked themes based on the frequency with which they were identified; it is reasonable to

suggest themes with the higher scores were identified by the greatest number of expert participants as being influential. In turn, this provides service providers and funders with an indication about which issues most require modification to improve the implementation of a program, and they can allocate their resources accordingly. Table 3, for example, shows that of the barriers identified, theme 1 (technical, logistical and referral challenges) is easily the issue that is identified by the greatest number of experts as a barrier (scoring -11) and could therefore be addressed as an issue requiring immediate modification to streamline program implementation.

Third, in addition to generating a frequency ranked list of barriers and enablers, the data presented in Table 2 can also help service providers and policymakers determine the experts who are likely to be most effectively engaged in addressing barriers. Table 2, for example, shows that the intermediaries group identified five of the six barriers, while the DCJ group identified none, which suggests that intermediaries are most likely to be usefully engaged in co-designing solutions to the identified barriers.

## Discussion

### Summary of findings

Using the modified scoring system proposed by the CFIR-f, three enablers and six barriers to implementing FFT-CW® and MST-CAN were identified. These themes were exactly the same as the themes identified using the original CFIR scoring system, which provides some evidence of concurrent validity for the CFIR-f given the original CFIR scoring mechanism can be regarded as the current gold standard measure. In contrast to the original CFIR scoring method, the CFIR-f allocated a continuous score to each theme. This generated a list of themes, ranked by the frequency with which each theme was identified by a range of experts involved in the delivery of the program. Synthesising the enablers and barriers in this way provides policymakers with an objective method for prioritising their decisions about how the implementation of a program might be most effectively and efficiently improved. In addition, the CFIR-f was able to identify the specific group of experts who are most likely to be usefully engaged in co-designing solutions to the identified barriers. Finally, the themes were able to be mapped to 65 of the 67 constructs from the latest adaptation of the CFIR (CFIR 2.0) [15].

### Comparisons to previous literature

The present study aims to enhance both the reliability with which themes are classified as enablers or barriers and make greater use of the implied consensus among expert participants, by allocating continuous scores to the themes that they identify. In terms of reliability, previous CFIR research relied on potentially subjective phrases, such as "in passing" or at a "high-level", to allocate a score to each of the identified enablers and barriers [7, 24], whereas this study allocates a score to each identified enabler and barrier based on the number of expert participants who identified it, which obviates the reliance of the potentially subjective phrases. In terms of frequency, previous CFIR research allocated a score from +2 to -2 for all themes, whereas this study only considered themes that were identified by at least 50% of expert participants as an enabler or barrier (to focus on the most frequently identified themes), and allocated a score based on the number of expert participants who identified each enabler and barrier to generate ranked themes based on a continuous, rather than ordinal, scale. This proposed frequency-based scoring approach meant that barriers, in particular, could be ranked and subsequently actioned based on precisely the number of participants who viewed them as inhibiting the implementation of a program.

To date, there has only been one published study employing CFIR in the child maltreatment field [4]. That study focused on the implementation of one of the programs examined in the current study (FFT-CW®) in another Australian state (Victoria). The authors of that study followed the original CFIR approach and developed interview questions in accordance with the five CFIR domains, whereas the questions used in the current study were instead developed for the purposes of the broader research evaluation of the FFT-CW® and MST-CAN programs in NSW (see S1 Appendix). Despite the differences in how the CFIR was utilised between that previous study [4] and this one, many similar implementation barriers were identified. These encompassed therapists feeling de-skilled (indicated here in the barrier, "workforce, staffing and resourcing issues"), lack of organisational readiness ("technical, logistical and referral challenges") and tensions between program and staff ("relationship with service providers and other stakeholders"). These similarities suggest that both approaches are appropriate for identifying implementation barriers.

## Methodological considerations

Greater utilisation of the number of expert participants who identified barriers to the implementation of FFT-CW® and MST-CAN allows service providers and policymakers to objectively address them using the more frequency-based approach proposed in this study. For example, addressing technical, logistical and referral challenges (allocated a score of -11) prior to the clarification and amendment of the eligibility criteria of the programs (allocated a score of -3) is likely to achieve greater improvements because there is more expert agreement about its negative influence on program implementation. The proposed CFIR-f would likely improve the reliability with which enablers and barriers to implementation are identified because it has less vulnerability to inter- and intra-rater errors related to inconsistencies in how phrases such as "in passing' and "high-level without examples" are interpreted by different evaluators or the same evaluators in different settings. In addition, the CFIR-f's greater emphasis on the frequency with which all experts identify themes would protect against the possibility of researchers unintentionally giving undue influence to certain themes in which they have tacit interest or influence [6, 12].

The accuracy with which enablers and barriers were identified by the CFIR-f also seems reasonable: the same enablers and barriers were identified by both the CFIR-f and the original CFIR scoring methods, and the order to which they were allocated was similar. Among the barriers, for example, the technical, logistical and referral challenges was ranked as the highest barrier using both the original scoring system and the CFIR-f (i.e., -6 and -11, respectively).

Furthermore, the relative advantage of the CFIR-f is that it generates a frequency-based list of enablers and barriers for policymakers to action based on continuous (not ordinal) scores from the perspectives of key stakeholders who have been actively engaged in the delivery of a program. In this study, for example, it is reasonable to conclude that technical, logistical and referral challenges [score -11] were identified about three times more frequently as a barrier to implementation than the need for clarification and amendment of the eligibility criteria of the programs ([score -3]). Using the original CFIR framework, both these themes would be allocated the same score of -2 for each expert group who identified them as barriers, irrespective of whether two or more stakeholders in each group articulated these themes as such. Thus, the original CFIR framework does not provide policymakers with sufficiently actionable information, that is, information that would be useful in assisting their decision-making about which of these barriers to prioritise to achieve relatively greater improvement in the ongoing implementation of a program.

In the real-world context of policymakers and service providers having limited resources and time, the continuous scores generated by the CFIR-f enhances the pragmatic usefulness of

the results of an implementation evaluation. Specifically, it generates information that originates directly from the experiences of key stakeholders who have been engaged in the implementation of a program, it is less susceptible to evaluator bias, and it presents a list of enablers and barriers that are ranked by the frequency with which they were identified. Generating and presenting information in this way aims to make that information more actionable for policymakers and service providers in determining the order in which barriers should be addressed, and enablers should be leveraged, to optimally improve implementation.

Despite using the 50% cut-off threshold for identifying themes (step 2), and the allocation of scores based on the number of participants who identified themes as enablers or barriers (step 3), the proposed CFIR-f scoring method could most likely be improved by further clarifying the criteria for how themes are classified and scored as neutral (0). As for the original CFIR, the CFIR-f approach still relies on the interpretation and judgement of researchers (e.g., to identify whether a theme is perceived by participants as 'indifferent' or 'unclear'). However, specifying that most participants are required to identify relevant themes in neutral terms, or that an equal number of participants are required to identify a specific theme as either an enabler or barrier to implementation, may provide the CFIR-f with a greater level of objectivity and standardisation, relative to the CFIR.

Another consequence of using the 50% cut-off threshold for identifying themes was that none of the themes identified by the DCJ participants were classified as enablers or barriers (because less than 50% of the DCJ group identified these themes). Although this suggests that there was less consensus regarding implementation issues among DCJ participants, relative to the other three policy and program expert groups, it may also highlight that further consideration could be given to how best to determine the level at which this cut-off threshold is set, including the idea that it might vary according to the circumstances of each evaluation. Alternatively, the cut-off threshold for identifying themes could be determined by using different, or more than one, criteria to reduce the reliance on one frequency-based criterion. For example, the threshold could be defined as including all enablers and barriers that were identified by at least 50% of participants and the most frequently cited enabler and barrier from any group whose views are excluded by the 50% threshold.

In addition, this initial version of the CFIR-f only considers the extent to which each enabler and barrier influences the implementation of a program through the assumption that the more frequently identified themes may be more influential. Future research could test the validity of this assumption. While the CFIR does not use this frequency-based approach (e.g., it only allocates a score of +2 or -2 to any theme identified by at least two participants), it does attempt to capture how strongly different enablers and barriers are likely to have influenced implementation by using complementary indicators, such as the level of agreement among participants, strength of language and the use of concrete examples [7]. However, a limitation of these indicators, and a key reason for the development of the CFIR-f, is their reliance on the evaluation team to interpret concepts such as "level of agreement" and "strength of language". Moreover, these indicators ultimately rely on the same type of assumption as the CFIR-f's frequency-based method in interpreting their pragmatic meaning: the use of "concrete examples" may reflect the relative simplicity of articulating a particular enabler or barrier, for instance, rather than the perceived strength of its influence on the implementation of a program.

A potential solution that could be tested in future research is the adoption of a matrix approach that incorporates both the frequency-based process of the CFIR-f and a less subjective method of capturing the relative impact of different enablers and barriers on the implementation of a program. The nominal group technique (NGT) method [25], for example, is a consensus-based approach to identifying priorities that the authors used in the evaluation of the FFT-CW® and MST-CAN programs in NSW [18, 19]. It may be possible to combine the

thematic analysis approach in this study with the NGT method in either a two-step process (e.g., semi-structured interviews and thematic analysis, followed by the ranking of themes using NGT in a subsequent session) or a one-step process if it becomes possible to harness artificial intelligence (AI) technology to perform thematic analysis with sufficient rapidity and accuracy.

## Implications for policy and practice

Using a standardised scoring and conceptual framework to assess the implementation of programs, such as the CFIR or the CFIR-f, should be routinely incorporated in the evaluations of public health programs to ensure there is standardisation in the way program enablers and barriers are identified. Solutions to barriers identified by expert participants (e.g., model purveyors and intermediaries) can then be explored using focus groups, co-design workshops or forums with a range of relevant stakeholders. Although both the CFIR and the CFIR-f are appropriate, the key implication of the latter for policy and practice is that it engenders a list of barriers ranked by frequency, which service providers and policymakers can use to inform their decisions about issues for action. Making the results of program implementation more actionable makes it easier for policymakers and funders to justify their decisions based on the ratings of independent experts.

Despite the CFIR-f being developed and piloted in the child maltreatment space, it is readily applicable to other programs delivered in education, workplace, legal or social services. It can also accommodate for iterative changes made to the constructs encompassed by the CFIR, such as the latest adaptations of CFIR 2.0 [15] which were incorporated in the present study, or the proposition to include real and perceived implementation and program outcomes, as well as outcome moderators, in future [26]. With enough applications of the CFIR-f, it is likely that it could be further refined to improve its practical usefulness in ensuring that barriers to program implementation, in particular, are identified as early as possible and modified to ensure they are maximally tailored to meeting the needs of their target populations. Integrating the views of other key participants involved in, or impacted by, the delivery of these programs into the CFIR-f scoring system, for example, may be warranted in future iterations. If the current study were to be replicated, this may include interviews with families who have had varying levels of engagement with the FFT-CW® and MST-CAN programs (i.e., families who exited early from, or who completed either program in full).

## Conclusions

This paper proposes a more frequency-based approach to CFIR's scoring process (the CFIR-f). The CFIR-f identified the same barriers and enablers as the original CFIR scoring process. By specifically utilising the frequency with which these barriers and enablers are identified, the CFIR-f engenders a list of ranked themes that service providers and policymakers can use to inform their decisions about program modification and implementation.

## Supporting information

**S1 Table. Key ideas of themes identified as barriers, enablers or neither from semi-structured interviews.**
(PDF)

**S2 Table. Definitions of CFIR domains and constructs.**
(PDF)

**S1 Appendix. Evaluation questions to assess adaptation and implementation of FFT-CW®
and MST-CAN in NSW, Australia.**
(PDF)

## Acknowledgments

The research team acknowledges and thanks all policy experts and service providers who participated in data collection for this study. We acknowledge the contribution of Dr. Paula Jops and Dr. Hueiming Liu to conducting interviews with some of the policy experts in this study.

## Author Contributions

**Conceptualization:** George Economidis, Anne-Marie Eades, Anthony Shakeshaft.

**Data curation:** George Economidis, Sara Farnbach.

**Formal analysis:** George Economidis, Sara Farnbach.

**Funding acquisition:** Anthony Shakeshaft.

**Investigation:** George Economidis, Sara Farnbach.

**Methodology:** George Economidis, Anne-Marie Eades.

**Project administration:** George Economidis, Anthony Shakeshaft.

**Resources:** Anthony Shakeshaft.

**Supervision:** Anthony Shakeshaft, Sara Farnbach.

**Validation:** Anne-Marie Eades.

**Writing – original draft:** George Economidis, Anthony Shakeshaft.

**Writing – review & editing:** George Economidis, Anne-Marie Eades, Anthony Shakeshaft,
Sara Farnbach.

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
