## [Decision Letter · Decision Letter 0]

18 Sep 2023

PONE-D-23-20754Improving the pragmatic usefulness of the scoring matrix for the Consolidated Framework for Implementation Research (CFIR). A proposal for a more frequency-based approach: the CFIR-f.PLOS ONE

Dear Dr. Economidis,

Thank you for submitting your manuscript to PLOS ONE. After careful consideration, we feel that it has merit but does not fully meet PLOS ONE’s publication criteria as it currently stands. Therefore, we invite you to submit a revised version of the manuscript that addresses the points raised during the review process.

ACADEMIC EDITOR: Thank you for this submission. It is clear and well written. Please respond to the points raised by the reviewer point wise. Thank you 

We look forward to receiving your revised manuscript.

Kind regards,

Dorothy Lall

Academic Editor

PLOS ONE

“Anthony Shakeshaft was the lead investigator on a competitive tender (FACS.17.266) awarded by the NSW Government Department of Family and Community Services (now Department of Communities and Justice) to evaluate the Functional Family Therapy – Child Welfare [FFT-CW®] and Multisystemic Therapy for Child Abuse and Neglect [MST-CAN] programs in New South Wales, Australia (2018-2020). George Economidis was employed part-time at the National Drug and Alcohol Research Centre as a Project Coordinator for the FFT-CW® and MST-CAN program evaluation from May 2018 to August 2020.”

“The research team acknowledges and thanks all policy experts and service providers who participated in data collection for this study. We acknowledge the contribution of Dr. Paula Jops and Dr. Hueiming Liu to conducting interviews with some of the policy experts in this study. NDARC receives funding from the Australian Government’s Department of Health.”

“GE was supported by an Australian Government Research Training Program (RTP) Scholarship via the University of New South Wales (UNSW), Sydney, Australia, and a Higher Degree Research scholarship from the National Drug and Alcohol Research Centre (NDARC), UNSW, Sydney. The evaluation of the MST-CAN program in New South Wales, Australia (2018-2020) was funded via a competitive tender process (FACS.17.266) by the NSW Government Department of Family and Community Services (now Department of Communities and Justice). Their Futures Matter and Department of Communities and Justice staff provided feedback on the evaluation plan, participated in interviews as key stakeholders in this evaluation, and reviewed the manuscript.”

Reviewers' comments:

Reviewer's Responses to Questions

**Comments to the Author**

1. Is the manuscript technically sound, and do the data support the conclusions?

Reviewer #1: Partly

2. Has the statistical analysis been performed appropriately and rigorously? 

Reviewer #1: N/A

3. Have the authors made all data underlying the findings in their manuscript fully available?

Reviewer #1: Yes

4. Is the manuscript presented in an intelligible fashion and written in standard English?

Reviewer #1: Yes

5. Review Comments to the Author

Reviewer #1: Thank you for the opportunity to review this paper, “Improving the pragmatic usefulness of the scoring matrix for the Consolidated Framework for Implementation Research (CFIR). A proposal for a more frequency-based approach: the CFIR-f.”

This paper reports and proposes a pragmatic, structured, frequency-based approach to applying the Consolidated Framework for Implementation Research +2/-2 scoring system. Given the number of domains, constructs and subconstructs (n=67) that the updated CFIR contains, innovative and pragmatic ways to analyse, and interpret, data obtained from applying the updated CFIR (and implementation theories, models and frameworks in general) is important. At the same time the authors identify several valid issues associated with previously published criteria to support researchers and practitioners apply the +2/-2 scoring system.

Specific questions and critiques follow in the order that they appear in the manuscript.

Abstract

Line 61: The CFIR-f engenders a list of themes, ranked by key experts according to their implied importance, to inform policymakers and services about priorities for action. I have serious concerns regarding this conclusion, is it not more accurate to conclude that the proposed scoring system generates a list of barriers and facilitators, ranked by frequency, rather than ‘according to their implied importance’.

Background

Line 67, missing references: Programs that simultaneously intervene at the individual, community and societal level are increasingly being utilised, both to optimise their effectiveness and improve the sustainability with which they are integrated into real-world service delivery.

Line 72: Identifying the enablers of, and barriers to, the delivery of multi-level programs at an early stage of their implementation is especially important because it is a particularly volatile stage in the program delivery process [1, 2]. I think it is important to add that identifying implementation determinants at an early stage can help inform implementation efforts (e.g., adaptations and modifications to the intervention and/or implementation strategy).

Line 77, missing reference: One such framework is the Consolidated Framework for Implementation Research (CFIR).

Line 87: In brief, implementation evaluators would typically convene a group of people involved in the delivery of a program and, using an established qualitative method (e.g., semi-structured interviews), ask a series of questions that align with some or all of CFIR’s constructs. This is one way CFIR can be applied, but I’m not convinced it can be described as the ‘typical’ way. I suggest the authors read/cite the following paper and provide a more accurate summary of how CFIR can has been used: Kirk, M.A., Kelley, C., Yankey, N. et al. A systematic review of the use of the Consolidated Framework for Implementation Research. Implementation Sci 11, 72 (2015). https://doi.org/10.1186/s13012-016-0437-z

Line 96: The answers to each of these questions would then be analysed using a standardised thematic analysis, and each theme is allocated a score by the evaluators ranging from +2 to -2: themes rated +1 or +2 are classified as enablers; themes rated -1 or -2 are classified as barriers; and themes rated as 0 are classified as being neither an enabler nor a barrier to implementation [4, 12]. Related to the previous point, I think it needs to be made clear that the +2/-2scoring system is frequently applied.

Line 101: I suggest rewording ‘actual barriers and enables’ to ‘more influential barriers and enablers’: The more extreme scores (i.e., +2 and -2) are interpreted as actual barriers and enablers, while +1 and -1 are interpreted as being generally influential. The word ‘actual’ implies that barriers and enables scored +1 or -1 are not ‘actual’ barriers and enablers, which is not the cases.

Methods

Table 1 is very useful and clear.

Results

Table 2 title add 'and neither enablers or barriers’. Please make it clear in the column heading, what the number in brackets in the first column represent. E.g. ‘Technical, logistical and referral challenges (n=19)’. This is not obvious.

Line 224: The logic is to provide services, funders and policymakers with a list of barriers and enablers that are weighted according to their importance, as determined by the frequency with which they are mentioned by the experts participating in each group. I disagree with the following part of this statement: ‘a list of barriers and enablers that are weighted according to their importance’. I strongly suggest changing to ‘a list of barriers and enablers by the frequency with which they are mentioned…’ This issue arises throughout the paper, please review and amend each statement accordingly (e.g., Line 387: the CFIR-f allocated a continuous score to each theme which can be interpreted as a list of themes prioritised by experts). The themes were not ‘prioritised’ by experts.

Discussion

Line 443. A major limitation to the CFIR-f approach is that identified barriers and enablers are ranked and ‘prioritised’ based solely on frequency, rather than frequency AND strength of influence on implementation. I don’t think the authors go far enough in highlighting and discussing the implications of this limitation. I think it would be useful to compare this approach to the original CFIR scoring approach, that determines strength of influence on implementation by considering a number of factors, including level of agreement among participants, strength of language, and use of concrete examples. Would a matrix approach, considering strength and frequency be the most desirable scoring approach? Perhaps this is something to highlight that should be explored in future research.

Line 457. I disagree that it is reasonable to conclude ‘that technical, logistical and referral challenges [score -11] is regarded by experts as being about three times more problematic than the need for clarification and amendment of the eligibility criteria of the programs ([score -3]).’ At best, the authors can state that technical, logistical and referral challenges were identified about 3 times more frequently as a barrier to implementation than the needs for clarification and amendment of the eligibility criteria of the programs. Frequency and the degree to which a barrier is ‘problematic’ are different.

How is the CFIR-f more pragmatic than the CRIF approach? It would be useful for the authors to define what they mean by ‘pragmatic usefulness’.

6. PLOS authors have the option to publish the peer review history of their article (what does this mean?). If published, this will include your full peer review and any attached files.

Reviewer #1: **Yes: **Louise Hull

---

## [Author Response · Author response to Decision Letter 0]

15 Oct 2023

Please see attached 'Response to Reviewers' uploaded file articulating my detailed responses to reviewer/editor comments. Thank you.

---

## [Editor Report · Decision Letter 1]

16 Nov 2023

Improving the pragmatic usefulness of the scoring matrix for the Consolidated Framework for Implementation Research (CFIR). A proposal for a more frequency-based approach: the CFIR-f.

PONE-D-23-20754R1

Dear Dr. Economidis,

We’re pleased to inform you that your manuscript has been judged scientifically suitable for publication and will be formally accepted for publication once it meets all outstanding technical requirements.

Kind regards,

Dorothy Lall

Academic Editor

PLOS ONE

Additional Editor Comments (optional):

Thank you all comments have been addressed and the manuscript has improved.
---

## [Editor Report · Acceptance letter]

21 Nov 2023

PONE-D-23-20754R1 

Improving the pragmatic usefulness of the scoring matrix for the Consolidated Framework for Implementation Research (CFIR). A proposal for a more frequency-based approach: the CFIR-f. 

Dear Dr. Economidis:

I'm pleased to inform you that your manuscript has been deemed suitable for publication in PLOS ONE. Congratulations! Your manuscript is now with our production department. 

Kind regards, 

on behalf of

Dr. Dorothy Lall 

Academic Editor

PLOS ONE